# OpenReview forum: "Leak@$k$: Unlearning Does Not Make LLMs Forget Under Probabilistic Decoding"
_ICLR.cc/2026/Conference — Submitted to ICLR 2026_

### Official Review · Reviewer_VUzu · 2025-10-20

**Soundness:** 2
**Presentation:** 3
**Contribution:** 2
**Rating:** 2
**Confidence:** 5

**Summary:**

This paper argues that current LLM unlearning methods only appear effective when evaluated using greedy (deterministic) decoding, but actually fail under probabilistic decoding (e.g., using temperature and top-p sampling). To expose this hidden leakage, the authors propose Leak@k, a meta-metric that measures the likelihood that sensitive or "forgotten" information reappears in at least one of k generated responses. They conduct a large-scale evaluation across three major unlearning benchmarks (TOFU, MUSE, WMDP) and show that as the number of generations increases, leakage rises significantly across all existing unlearning methods, revealing that current techniques do not guarantee reliable forgetting in practical usage settings.

**Strengths:**

1. The motivation for introducing a new metric for unlearning evaluation is well-founded.

2. The paper is well-written and easy to follow.

3. The experimental section is comprehensive.

**Weaknesses:**

1. The proposed metric Leak@K requires the unlearned LLM to generate multiple responses for each prompt, which significantly increases the computational cost of evaluation.

2. The idea of using probabilistic decoding to evaluate model memorization is not new, as it has been explored in prior works [1][2].

3. The overall contribution of the paper appears limited, as it primarily introduces a new evaluation metric for unlearning without proposing a corresponding unlearning method or broader framework.



[1] Quantifying Memorization Across Neural Language Models. ICLR 2023.
[2] A Probabilistic Perspective on Unlearning and Alignment for Large Language Models. ICLR 2025.

**Questions:**

please see weakness.

---

> ### Author Response · Authors · 2025-11-20
>
> > The proposed metric Leak@K requires the unlearned LLM to generate multiple responses for each prompt, which significantly increases the computational cost of evaluation.
>
> **Response:** We respectfully disagree with the reviewer’s concern regarding computational cost. First, leak@k is **computationally mild** (details in table 1). In fact, one can view leak@k as a kind of "adversarial" attack to the unlearned model, and it adheres to the established principle in adversarial evaluation: allocating additional compute to more reliably surface residual safety failures in LLMs [8]. Importantly, it remains substantially **more efficient than existing adversarial methodologies**: targeted relearning attacks require hours of additional fine-tuning [9], and jailbreak attacks such as Greedy Coordinate Gradient-based Search (GCG) [10] may require more than a day of compute. By contrast, leak@k relies solely on lightweight sampling, making its computational overhead modest and fully justified.
>
> |  Benchmark  | Time for Generation | Time for Evaluation RS/ES |
> |:-----------:|:-------------------:|:------------------------------:|
> | TOFU |        10min        |             30min           |
> |    MUSE     |        1h             | 15min                               |
> |    WMDP     |        1h             |       15min    |
>
>
> **Table 1.** Across all three benchmarks, we generate $n=200$ model responses per prompt. The TOFU and WMDP datasets contain 400 and 200 prompts, respectively. “Generation time” is the total time to produce all responses for the full prompt set. “Evaluation time” is the time to compute ES/RS leakage scores for all generated responses. All generation and ES/RS Evluation steps are done by using one or two NVIDIA A40 GPUs.
>
> Second, we would like to clarify that evaluating leak@k does **not introduce any meaningful real-world concern**; instead, it **offers several benefits** for two key reasons.
>
>
> (1) To reliably assess unlearning robustness, one must capture **worst-case** resurfacing, which necessarily requires multiple generations—our leak@k reflects realistic deployment, where **adversaries** can **repeatedly** query the model.
>
> (2) In modern LLM evaluation, multi-sample evaluation is standard practice. **Pass@k** is widely used in **SWE**, **HumanEval**, and **GSM8K** [11, 12] precisely because multiple generations are essential for capturing the output distribution and obtaining stable performance estimates. Our metric leak@k follows the **same** principle.
>
> In summary, leak@k is **far from the computational burden** the reviewer suggests; on the contrary, practitioners are likely to welcome an evaluation metric that **delivers more reliable unlearning assessment** while requiring only lightweight computation. This makes leak@k **a practical and valuable metric for guiding the design of future unlearning algorithms.**
>
>
>
> > The idea of using probabilistic decoding to evaluate model memorization is not new, as it has been explored in prior works [1*][2*].
> > [1*] Carlini, N., Ippolito, D., Jagielski, M., Lee, K., Tramer, F., & Zhang, C. Quantifying memorization across neural language models, 2022.
> [2*] Scholten, Y., Günnemann, S., & Schwinn, L. A probabilistic perspective on unlearning and alignment for large language models, 2024.
>
> **Response:**
>
> We note that [1*] uses beam search, **which is deterministic, not probabilistic**; it deterministically computes log-probabilities and selects the top-k candidates at each decoding step as stated in [13] which develops the original beamsearch algorithm. Given this, [1*] is clearly irrelevant to our probabilistic setting.
>
>
> Although [2*] explores probabilistic decoding, the proposed approach remains limited: (1) it provides only token-level, distributional guarantees and does not assess whether sampled outputs still reveal the semantic meaning of the forgotten knowledge; (2) Their analysis effectively measures the leakage probability of a single sample. They do not model what happens when a user or adversary draws multiple samples, where even very small leakage rates can grow quickly. Our metric is the first to evaluate the robustness of unlearning under real-world deployment.

---

> ### Author Response · Authors · 2025-11-20
>
> > The overall contribution of the paper appears limited, as it primarily introduces a new evaluation metric for unlearning without proposing a corresponding unlearning method or broader framework.
>
> **Response:**
> We respectfully disagree with the reviewer that our work does not propose any corresponding unlearning. We **indeed propose a new algorithm**, NPO-Fix, which is mentioned on page 9 and fully elaborated in the appendix. In the following, we provide a brief description of the NPO-Fix method and our new results (developed after submission of the first version of the paper) for this algorithm. We will include these new results with its full details in the revision.
>
>
> Our proposed NPO-Fix includes a dataset-enhancement stage that filters generations exhibiting leakage and augments the training corpus with these instances. We have further tuned hyperparameters for NPO-Fix, and experiments show that incorporating generation-aware training during unlearning mitigates leakage to a meaningful extent and **achieves substantially better performance than existing methods**. We view this as a promising direction for future research.
>
>
> | Method   | 1    | 2    | 4    | 8    | 16   | 32   | 64   | 128  |
> | -------- | ---- | ---- | ---- | ---- | ---- | ---- | ---- | ---- |
> | Retrain  | 16.9 | 23.6 | 30.6 | 37.6 | 44.1 | 50.1 | 55.7 | 61.0 |
> | Original |  28.7    |   40.5   |   52.4   | 63.1     | 72.1     | 79.4     |  84.9    | 89.1     |
> |   NPO       | 20.4     | 29.2     | 38.4     |  47.3    |  55.4    |  62.5    | 68.7     |  74.3    |
> | NPO-Fix  | **0.4**  | **0.8**  | **1.5**  | **2.7**  | **4.3**  | **6.5**  | **9.4**  | **13.4** |
>
> [8] Łucki, J., Wei, B., Huang, Y., Henderson, P., Tramèr, F., & Rando, J.. An adversarial perspective on machine unlearning for AI safety. (2024)
>
> [9] Hu, S., Fu, Y., Wu, S., & Smith, V.. Jogging the memory of unlearned models through targeted relearning attacks. ICML 2024 Workshop on Foundation Models in the Wild.
>
> [10] Zou, A., Wang, Z., Carlini, N., Nasr, M., Kolter, J. Z., & Fredrikson, M.. Universal and transferable adversarial attacks on aligned language models. (2023)
>
> [11] Chen, M. Evaluating large language models trained on code, 2021.
>
> [12] Brown, B., Juravsky, J., Ehrlich, R., Clark, R., Le, Q. V., Ré, C., & Mirhoseini, A. Large language monkeys: Scaling inference compute with repeated sampling, 2024.
>
> [13] Decoding Strategies in Large Language Models. https://huggingface.co/blog/mlabonne/decoding-strategies

---

> > ### Comment · Reviewer_VUzu · 2025-11-24
> >
> > Thanks for your response. But I am still not convinced by the rebuttal.
> >
> > (1) Regarding results in Table 1, I don't think it is efficient.
> >
> > (2) As demonstrated in the rebuttal, the novelty beyond (A Probabilistic Perspective on Unlearning and Alignment for Large Language Models. ICLR 2025.) is trivial.
> >
> > (3) The introduction concludes without mentioning that the paper has proposed a new algorithm, and it is confusing to introduce the new algorithm on page 9 (the last page of main content). The structure of this paper can be significantly improved.

---

> ### Author Response · Authors · 2025-11-27
>
> We thank the reviewer for the comments. We believe there may still be some misunderstandings regarding our contributions, and we would like to clarify the following points:
>
> (1) **Multi-sample** evaluation is a **standard** paradigm in the LLM community [14, 15] where efficiency is generally not considered as a constraint. We note that prior works have utilized up to **1,000,000** generations to assess coding capabilities [14] or sampled **40 diverse reasoning paths** to enhance consistency [15]. Hence, the time spent on generations/evaluations is acceptable in exchange for a robust performance estimation. Accordingly, **our objective is not inference efficiency**, but rather to propose a meta-metric that evaluates the robustness of machine unlearning under real-world LLM settings. Our contribution is to reveal a critical, previously unobserved failure mode: *unlearning methods that appear successful under greedy decoding consistently fail when subjected to multi-sample generation*.
>
>
> (2) Our work is **fundamentally different** from [2*]. That paper *does not identify the key phenomenon where probabilistic decoding exposes substantially more leakage than greedy decoding across all major unlearning methods and benchmarks*. Our paper is the first to introduce a meta-metric (leak@k) that directly measures information leakage under probabilistic decoding. Moreover, the unlearning method proposed in [2*] (**NPO+ENT**) *performs well under their metric* but **fails** under **leak@k**, further demonstrating that our metric captures failure modes that prior evaluations completely miss.
>
>
> (3) As noted above, our **primary contribution** is to **introduce a metric** that captures information leakage of unlearned models under realistic deployment settings. The goal of the paper is an evaluation framework, not to center a new unlearning algorithm. The inclusion of a new method (NPO-Fix) is supplementary and not intended as the main contribution. While **NPO-Fix** achieves the **SOTA performance** on the TOFU benchmark for the leak@k metric (you can check our previous response).
>
> [14] Li, Y., Choi, D., Chung, J., Kushman, N., Schrittwieser, J., Leblond, R., ... & Vinyals, O.. Competition-level code generation with alphacode. Science, 2022.
>
> [15] Wang, X., Wei, J., Schuurmans, D., Le, Q., Chi, E., Narang, S., ... & Zhou, D. Self-consistency improves chain of thought reasoning in language models. ICLR 2023.

---

### Official Review · Reviewer_RrXa · 2025-10-31

**Soundness:** 3
**Presentation:** 3
**Contribution:** 3
**Rating:** 6
**Confidence:** 3

**Summary:**

This submission points out a critical problem that existing large language model (LLMs) unlearning methods do not really make LLMs forget the knowledge. The greedy coding setting does not really reflect the unlearning methods’ performance in real-world settings. This paper proposes a metric named leak@k to facilitate the evaluations of LLM unleaning.

**Strengths:**

This paper has the following strengths:
- This paper is clearly written and easy to read.
- This paper identifies a practical and important limitation in evaluating large language model unlearning.
- The proposed metric demonstrates effectiveness in facilitating LLM unlearning.

**Weaknesses:**

This paper has the following weaknesses:
- The evaluated models and methods are limited.
- There is a lack of insight into why these unlearning methods fail to perform effectively outside the greedy setting.

**Questions:**

I have the following questions/suggestions:
- The reference formatting appears to be incorrect. It would be better for citations to follow the Name (Year) or (Name, Year) styles as appropriate to the context, rather than using only the Name (Year) format.
- Evaluate on more model sizes like 13B scale.

---

> ### Author Response · Authors · 2025-11-30
>
> > The evaluated models and methods are limited.
>
> **Response:** We thank the reviewer for the feedback. We thank the reviewer for the feedback. In the revision, we substantially expanded the set of evaluated models and methods for each benchmark. Below, we provide a summary of the newly added results, with full experimental details reported in the revision.
>
> **TOFU.** We implemented the method from [4*], **LoUK**, on the TOFU benchmark using  LLaMA-3.2-1B. **Similar** to **all** other unlearned models, we observe that this method exhibits **clear** information **leakage** under relatively high-randomness decoding strategies. We provide heatmaps for each configuration in **Fig. 2**, **page 8** and **Fig. A1**, **page 19**. Below, we report some of the obtained results for **LoUK** model on TOFU dataset using **ES** as the core metric.
>
> | (temp., top-$p$)   | 1    | 2    | 4    | 8    | 16   | 32   | 64   | 128  |
> | -------- | ---- | ---- | ---- | ---- | ---- | ---- | ---- | ---- |
> | $(0.0,0.0)$ (greedy decoding)  | 0.015 | 0.02 | 0.02 | 0.02| 0.02 | 0.02 | 0.02 | 0.02
> | $(0.2,0.2)$ |  0.03    |   0.03   |   0.03   | 0.03     | 0.03     | 0.03     |  0.03    | 0.03     |
> |   $(0.2,1.0)$       | 0.07     | 0.08     | 0.10     |  0.12    |  0.15    |  0.20    | 0.27     |  0.38   |
> | $(1.0,1.0)$  | 0.05 | 0.08  | 0.13  | 0.20  | 0.29  | 0.37  | 0.48  | 0.60 |
>
>
> We also conducted a new set of experiments on this dataset using a new core metric, **LLM-as-Judge (LJ)**, evaluated on all unlearned models under two sampling configurations: a low-randomness setting $(T,p)=(0.2,0.2)$ and a high-leakage setting $(T,p)=(1.0,1.0)$, reported in **Fig 3**, **page 8**. **Aligned** closely with the behavior observed using the **ES** metric in Fig. 2, page 8, the results obtained with **LJ** **confirm** that under high-randomness sampling, the unlearned model **exhibits** a clear risk of information leakage. In the following, we present the result of leak@k from some unlearned models using LJ metric under $(T,p)=(1.0,1.0)$ setting on TOFU dataset.
>
> | Method   | 1    | 2    | 4    | 8    | 16   | 32   | 64   | 128  |
> | -------- | ---- | ---- | ---- | ---- | ---- | ---- | ---- | ---- |
> | GradDiff  | 0.40 | 0.54 | 0.65 |0.74| 0.82 | 0.87| 0.92 | 0.95 |
> | NPO |  0.35    |   0.46   |   0.57   | 0.65     | 0.73     | 0.79    | 0.85   | 0.90    |
> |   BLUR-NPO       | 0.25    | 0.35  |0.45    |  0.40   |  0.62   |  0.69   | 0.74     |  0.79    |
> | LoUK  | 0.03  | 0.05  | 0.08  | 0.11  | 0.15  | 0.19  | 0.24  | 0.29 |
>
>
> **MUSE.** We implemented the proposed algorithm in [1*], **SURE**, using  LLaMA2-7B model and evaluated it on the MUSE benchmark with **RS** as the core metric. **Similar** to other models, we observe that this method also **fails** under our metric leak@k under relatively high-randomness decoding strategies. We provide heatmaps for each configuration in **Fig. 4**, **page 9** and **Fig. A3**, **page 20**. Below, we report some of the obtained results for **SURE** model.
>
>
> | (temp., top-$p$)   | 1    | 2    | 4    | 8    | 16   | 32   | 64   | 128  |
> | -------- | ---- | ---- | ---- | ---- | ---- | ---- | ---- | ---- |
> | $(0.0,0.0)$ (greedy decoding)  | 0.30 | 0.30 | 0.30 | 0.30 | 0.30 | 0.30 | 0.30 | 0.30 |
> | $(0.2,0.2)$ |  0.32    |   0.32   |   0.32   | 0.32     | 0.32     | 0.32    |  0.32    | 0.32     |
> |   $(0.2,1.0)$       | 0.33     | 0.43     | 0.54     |  0.64    |  0.72    |  0.80    | 0.86     |  0.90    |
> | $(1.0,1.0)$  | 0.25  | 0.37  | 0.50  | 0.62  | 0.73  | 0.82  | 0.89  | 0.94 |
>
>
>
> **WMDP.** We further extended our evaluation on WMDP by applying **LJ** to RMU model for two sampling configurations $(T,p)=(0.2,0.2)$ and $(T,p)=(1.0,1.0)$. **Similar** to the **Acc** core metric, RMU exhibits clear information leakage under high-randomness decoding setting $(T,p)=(1.0,1.0)$.
>
> | (temp., top-$p$)   | 1    | 2    | 4    | 8    | 16   | 32   | 64   | 128  |
> | -------- | ---- | ---- | ---- | ---- | ---- | ---- | ---- | ---- |
> | $(0.2,0.2)$  | 0.047 | 0.051 | 0.057 | 0.064| 0.071  | 0.080 | 0.091 | 0.108 |
> | $(1.0,1.0)$ |  0.037    |   0.052   |   0.068   | 0.086     | 0.107     | 0.127     |  0.145   | 0.162   |
>
>
> > There is a lack of insight into why these unlearning methods fail to perform effectively outside the greedy setting.
>
>
> **Response:**  We thank the reviewer for the comment. The failure of the greedy decoding stems from a fundamental mismatch between **how existing unlearning methods are trained** and **how LLMs are deployed**. Most unlearning objectives reduce the likelihood of undesired outputs along a **single** greedy decoding path (**most probable token**), but **do not remove** the **residual** probability mass assigned to the forgotten knowledge. As a result, while greedy decoding could suppress leakage, probabilistic sampling exposes remaining memorized information.

---

> > ### Author Response · Authors · 2025-11-30
> >
> > > The reference formatting appears to be incorrect. It would be better for citations to follow the Name (Year) or (Name, Year) styles as appropriate to the context, rather than using only the Name (Year) format.
> >
> > **Response:** We thank the reviewer for the comment. We fixed the formatting properly in the revision.
> >
> > > Evaluate on more model sizes like 13B scale.
> >
> > **Response:** We appreciate the suggestion. In principle, evaluating larger models such as 13B would be valuable. Although we are technically able to evaluate larger models (e.g., 13B), to the best of our knowledge, **no** 13B unlearned models currently exist for TOFU, MUSE, or WMDP due to **computational resource constraints**. We note that models of this size (up to 7B) are **widely used** in academic research [16, 17, 18, 19], making them well-suited for the scope and objectives of our study.
> >
> >
> > [16] Weijia Shi, Jaechan Lee, Yangsibo Huang, Sadhika Malladi, Jieyu Zhao, Ari Holtzman, Daogao Liu, Luke Zettlemoyer, Noah A Smith, and Chiyuan Zhang. Muse: Machine unlearning six-way evaluation for language models, 2024.
> >
> > [17] Nathaniel Li, Alexander Pan, Anjali Gopal, Summer Yue, Daniel Berrios, Alice Gatti, Justin D Li, Ann-Kathrin Dombrowski, Shashwat Goel, Long Phan, et al. The wmdp benchmark: Measuring and reducing malicious use with unlearning, 2024.
> >
> > [18]  Ruiqi Zhang, Licong Lin, Yu Bai, and Song Mei. Negative preference optimization: From catastrophic collapse to effective unlearning, 2024.
> >
> > [19] Pratyush Maini, Zhili Feng, Avi Schwarzschild, Zachary C. Lipton, and J. Zico Kolter. Tofu: A task of fictitious unlearning for llms, 2024.

---

### Official Review · Reviewer_A144 · 2025-11-01

**Soundness:** 1
**Presentation:** 3
**Contribution:** 1
**Rating:** 2
**Confidence:** 4

**Summary:**

The paper proposes leak@k, a novel meta-evaluation metric designed to measure the LLM unlearning efficacy by quantifying how often unwanted knowledge appears when generating under probabilistic decoding. Using this metric, the authors conduct a large-scale evaluation of unlearning efficacy across three widely used LLM unlearning benchmarks: TOFU, MUSE, and WMDP. The analysis reveals that, although existing unlearning methods appear effective under deterministic greedy decoding, sensitive information frequently resurfaces when standard sampling strategies are used. These findings demonstrate that current unlearning techniques commonly fail to achieve true forgetting in practice, highlighting the need for more reliable and principled approaches to LLM unlearning.

**Strengths:**

- [S1] **Interesting and relevant direction.** The paper addresses a timely and important issue in LLM research, the reliability of unlearning, and provides a new perspective by re-examining prior work under probabilistic decoding. This direction is valuable given the growing importance of safe and compliant model deployment.
- [S2] **Comprehensive experimental coverage.** The authors perform extensive empirical studies across multiple established benchmarks (TOFU, MUSE, and WMDP), offering a fairly broad assessment of existing unlearning techniques under different decoding conditions.
- [S3] **Clarity and structure.** The paper is generally well-written and easy to follow. The motivation, setup, and presentation of results are clear and logically connected.

**Weaknesses:**

- [W1] **Misalignment with the goal of unlearning.** The paper’s overall analysis appears misaligned with the conventional goal of unlearning, which is to make the unlearned model approximate the retain model, rather than simply avoid producing the correct answer. Accordingly, the Retrain curve in Figure 1 should be interpreted as a performance upper bound that effective unlearning methods should aim to approach, yet this interpretation is never discussed. In several cases (e.g., Tables 1, 7, and 8), even the retrained model generates the same “forgotten” answer, possibly due to hallucination or because the information legitimately exists in the retained data. Penalizing the unlearned model for such behavior seems conceptually inconsistent. This ambiguity raises a fundamental question of whether leak@k truly measures unlearning efficacy or merely captures aspects of refusal and alignment. Without a clearer definition of what constitutes a valid “leak,” the evaluation risks conflating unlearning failures with general response suppression or alignment artifacts.
- [W2] **Weak theoretical justification.** The theoretical section contributes little to supporting the main claims. The proof that the proposed estimator is unbiased is mathematically correct but does not meaningfully strengthen the argument for why leak@k is an appropriate or informative metric for unlearning. Unless the paper first establishes why leak@k captures the core desiderata of forgetting, the theoretical analysis feels disconnected from the empirical motivation. Furthermore, the remark that pass@k is a special case of leak@k is not clearly explained in terms of its practical significance for unlearning evaluation.
- [W3] **Metric choice and dependency on data format.** The evaluation relies on different similarity metrics (e.g., entailment score for TOFU, ROUGE for MUSE-News, mixed rubric for WMDP), and this choice appears strongly dependent on the specific format of the benchmark. This raises concerns about comparability and potential bias: for example, how short or long should answers be to justify using ROUGE vs. entailment score? What happens when the forget set includes paraphrased or reformatted variants of the same information? This dependence suggests that the evaluation pipeline may require dataset-specific manual tuning, reducing its generalizability and introducing potential evaluation bias.
- [W4] **Decoding parameter sensitivity.** The results show that increasing temperature and top-p generally increases leakage, but the paper does not clarify what decoding configurations are considered “standard” or “representative” for evaluating successful unlearning. For instance, is $T = 1.0$ chosen as an upper bound for theoretical reasons or empirical convenience? High temperature may also introduce randomness that confounds the distinction between resurfacing knowledge and random noise. Without clearer guidance or principled justification for which parameter setup is proper for evaluation, it is hard to interpret what constitutes a fair or meaningful evaluation under leak@k.

**Questions:**

- [Q1] The TOFU benchmark already measures the probability of generating paraphrased answers from the forget set, which implicitly accounts for uncertainty without relying on greedy decoding. How does leak@k differ conceptually from this evaluation protocol, and does it truly capture a distinct failure mode?
- [Q2] There are several instances where citations are formatted inconsistently (e.g., only the year is enclosed in parentheses). Please ensure that citations consistently include both author and year when appropriate.

---

> ### Author Response · Authors · 2025-11-20
>
> > **Misalignment with the goal of unlearning.** The paper’s overall analysis appears misaligned with the conventional goal of unlearning, which is to make the unlearned model approximate the retain model, rather than simply avoid producing the correct answer. Accordingly, the Retrain curve in Figure 1 should be interpreted as a performance upper bound that effective unlearning methods should aim to approach, yet this interpretation is never discussed. In several cases (e.g., Tables 1, 7, and 8), even the retrained model generates the same “forgotten” answer, possibly due to hallucination or because the information legitimately exists in the retained data. Penalizing the unlearned model for such behavior seems conceptually inconsistent. This ambiguity raises a fundamental question of whether leak@k truly measures unlearning efficacy or merely captures aspects of refusal and alignment. Without a clearer definition of what constitutes a valid “leak,” the evaluation risks conflating unlearning failures with general response suppression or alignment artifacts.
>
> **Response:** We respectfully disagree with the reviewer's assumption that approximating the retrain model is the upper bound in LLM unlearning, while it holds true for conventional machine unlearning. In LLM unlearning, the retrain model is not a valid ideal target: while not exposed to forget set, retrain model may still retain and resurface information even under greedy decoding (e.g., as shown for the MUSE benchmark [1,2]).  Hence, using it as the **ideal** behavior is conceptually **incorrect**. A successful unlearning method is instead defined by **low leakage on the forget set** and **high utility on the retain set**, regardless of how the retrain model behaves under either greedy or probabilistic decoding. Our paper focuses on this correct objective, rather than approximating the retrain model.
>
>
> > **Weak theoretical justification.** The theoretical section contributes little to supporting the main claims. The proof that the proposed estimator is unbiased is mathematically correct but does not meaningfully strengthen the argument for why leak@k is an appropriate or informative metric for unlearning. Unless the paper first establishes why leak@k captures the core desiderata of forgetting, the theoretical analysis feels disconnected from the empirical motivation. Furthermore, the remark that pass@k is a special case of leak@k is not clearly explained in terms of its practical significance for unlearning evaluation.
>
> **Response:** Our paper does not claim theoretical guarantees for unlearning, nor would such guarantees be meaningful in this context. As stated in the paper, leak@k is an **evaluation** meta-metric, not an unlearning algorithm, and the role of the theoretical section is appropriately limited to **providing unbiased estimators for this metric**. This is **standard** for evaluation-metric research [3,4,5,6]. In real-world settings such as copyright leakage, the risk is inherently probabilistic. Harm does not appear only when a model deterministically outputs a specific undesired information. It arises whenever the model has a good chance of reproducing sensitive content across its sampled generations. The remark that pass@k is a special case of leak@k is **simply a clarification** of our unbiased estimator meta-metric, not a claim of additional practical significance. This observation shows that leak@k generalizes an already widely used sampling-based metric and therefore fits naturally within existing evaluation practices.

---

> ### Author Response · Authors · 2025-11-20
>
> > **Metric choice and dependency on data format.** The evaluation relies on different similarity metrics (e.g., entailment score for TOFU, ROUGE for MUSE-News, mixed rubric for WMDP), and this choice appears strongly dependent on the specific format of the benchmark. This raises concerns about comparability and potential bias: for example, how short or long should answers be to justify using ROUGE vs. entailment score? What happens when the forget set includes paraphrased or reformatted variants of the same information? This dependence suggests that the evaluation pipeline may require dataset-specific manual tuning, reducing its generalizability and introducing potential evaluation bias.
>
> **Response:** We thank the reviewers for raising concerns regarding our metric choices. We would like to clarify that we did **not perform any manual tuning** of evaluation metrics. For each benchmark, we directly adopt the standard core metrics defined in the original works [1, 3, 7], and we never adjust or tune these metrics to reveal or amplify information leakage; all these metrics show information leakage in consistence.
> Regarding the reviewer’s concerns about generalizability and potential bias, we emphasize that leak@k is a meta-metric, which can be **seamlessly applied to any base metric that based on generations**. As long as one metric can reflect information leakage, it can be incorporated within the leak@k framework. Thus, leak@k inherits the validity of the base metric (such as Entailment Score, Rouge Score), providing a unified and generalizable way to quantify leakage under probabilistic multiple generations.
>
> > **Decoding parameter sensitivity.** The results show that increasing temperature and top-p generally increases leakage, but the paper does not clarify what decoding configurations are considered “standard” or “representative” for evaluating successful unlearning. For instance, is chosen as an upper bound for theoretical reasons or empirical convenience? High temperature may also introduce randomness that confounds the distinction between resurfacing knowledge and random noise. Without clearer guidance or principled justification for which parameter setup is proper for evaluation, it is hard to interpret what constitutes a fair or meaningful evaluation under leak@k.
>
> **Response:** Our work performs a **complete** and **systematic** sweep across all widely used decoding configurations, varying temperature and top-p over the **full** **standard** ranges. We do not **tune** or select any particular (temperature, top-p) configuration. *As shown in Figs. A2–A3, model utility remains stable across all settings, so increases in leak@k are due to true semantic resurfacing, not randomness.* Also, we want to clarify that all variability for probabilistic decoding comes from the LLM’s own distribution being flattened by temperature (shown in Eq. (1)), **not from any external random noise**; we disagree with the statement "confounds the distinction between resurfacing knowledge and random noise".
> $$
> P_i = \frac{e^{y_i / T}}{\sum_{k=1}^{n} e^{y_k / T}}
> \tag{1}
> $$
>
> where $y_i$ is the pre-softmax output (logit) of the LLM corresponding to the $i$-th token, and $P_i$ is the resulting probability of sampling that token under temperature $T$. Increasing the temperature $T$ flattens the distribution by reducing the relative differences between logits $y_i$, while decreasing $T$ sharpens it.

---

> ### Author Response · Authors · 2025-11-20
>
> > The TOFU benchmark already measures the probability of generating paraphrased answers from the forget set, which implicitly accounts for uncertainty without relying on greedy decoding. How does leak@k differ conceptually from this evaluation protocol, and does it truly capture a distinct failure mode?
>
> **Response:** We appreciate the reviewer for noticing the distinction between paraphrased probability and leak@k, but the two differ in both **methodology** and **effectiveness**.
> 1. **Methodologically**. Paraphrased probability evaluates the **likelihood of a single paraphrased answer** (see Eq. (2)) and does not involve generation[8], so it cannot reflect how an unlearned model behaves under probabilistic decoding or what its output distribution looks like.
> $$
> \text{Para. Prob.} = p\left( f\left( y^{\text{para}} \mid x \right) \right)
> \tag{2}
> $$
>
> 2. **Effectively**. Because paraphrased answers are **pre-generated and fixed**, they cannot approximate the full support of the model’s predictive distribution. In contrast, leak@k directly samples from the model’s probabilistic decoding process and thus provides a more reliable estimation of leakage under multiple generations.
>
> In summary, leak@k is a **completely different** metric to capture information leakage, which will have a **more reliable assessment** than paraphrased probability.
>
> > There are several instances where citations are formatted inconsistently (e.g., only the year is enclosed in parentheses). Please ensure that citations consistently include both author and year when appropriate.
>
> **Response:** We thank the reviewer for the comment. We address the citation formatting in the revised version of the paper.
>
> [1] Weijia Shi, Jaechan Lee, Yangsibo Huang, Sadhika Malladi, Jieyu Zhao, Ari Holtzman, Daogao Liu, Luke Zettlemoyer, Noah A Smith, and Chiyuan Zhang. Muse: Machine unlearning six-way evaluation for language models, 2024.
>
> [2] Chongyu Fan, Jiancheng Liu, Licong Lin, Jinghan Jia, Ruiqi Zhang, Song Mei, and Sijia Liu. Simplicity prevails: Rethinking negative preference optimization for llm unlearning, 2024.
>
> [3] Xiaojian Yuan, Tianyu Pang, Chao Du, Kejiang Chen, Weiming Zhang, and Min Lin. A closer look at machine unlearning for large language models, 2024.
>
> [4] Wang, Q., Han, B., Yang, P., Zhu, J., Liu, T., & Sugiyama, M. Towards effective evaluations and comparisons for llm unlearning methods, 2024.
>
> [5] Feng, Z., Xu, Y. E., Robey, A., Kirk, R., Davies, X., Gal, Y., ... & Kolter, J. Z. Existing Large Language Model Unlearning Evaluations Are Inconclusive, 2025.
>
> [6] Cho, S., Hwang, D., Sala, F., Hwang, S., Cho, K., & Cha, S. Reference-Specific Unlearning Metrics Can Hide the Truth: A Reality Check, 2025.
>
> [7] Nathaniel Li, Alexander Pan, Anjali Gopal, Summer Yue, Daniel Berrios, Alice Gatti, Justin D Li, Ann-Kathrin Dombrowski, Shashwat Goel, Long Phan, et al. The wmdp benchmark: Measuring and reducing malicious use with unlearning, 2024.
>
> [8] Dorna, V., Mekala, A., Zhao, W., McCallum, A., Lipton, Z. C., Kolter, J. Z., & Maini, P.. OpenUnlearning: Accelerating LLM Unlearning via Unified Benchmarking of Methods and Metrics，2025.

---

> ### Comment · Reviewer_A144 · 2025-11-28
>
> Thank you authors for your time and commitment made into the rebuttal. However, a few of my concerns still remain.
>
> ---
> > We respectfully disagree with the reviewer's assumption that approximating the retrain model is the upper bound in LLM unlearning, while it holds true for conventional machine unlearning. In LLM unlearning, the retrain model is not a valid ideal target: while not exposed to forget set, retrain model may still retain and resurface information even under greedy decoding (e.g., as shown for the MUSE benchmark [1,2]). Hence, using it as the ideal behavior is conceptually incorrect.
>
> **Response:** I appreciate the authors' perspective on this conceptual distinction. Nonetheless, I respectfully maintain that the retrained model remains the appropriate target for approximate unlearning, even with its mentioned limitations.
>
> The concern that retrained models may retain information from the forget set is well-taken. Nevertheless, **the cited MUSE benchmark [A], along with SimNPO [B] and TOFU [C], all adopt the retrained model as the gold standard** for evaluating unlearning methods (see, for example, the PrivLeak metric in [A] and Figure 1 in [B]). This consensus reflects the fundamental objective of **un**learning: to restore the model to a state and **undo** the effect of training on the forget set.
>
> I would argue that if the goal is to achieve behavior that deviates from that of the retrained model, then we are addressing a different problem. Such an objective would be more aligned with **model alignment or safety interventions** rather than unlearning. While I agree that measuring low leakage and high utility are essential metrics, in the context of unlearning these metrics should be evaluated in how well they approximate the retrained baseline, rather than other independent objectives.
>
> [A] [Shi et al., MUSE: Machine Unlearning Six-Way Evaluation for Language Models. ICLR 2025.](https://arxiv.org/abs/2407.06460)\
> [B] [Fan et al., Simplicity Prevails: Rethinking Negative Preference Optimization for LLM Unlearning. NeurIPS 2025.](https://arxiv.org/abs/2410.07163)\
> [C] [Maini et al., TOFU: A Task of Fictitious Unlearning for LLMs. COLM 2024.](https://arxiv.org/abs/2401.06121)
>
> ---
> > We would like to clarify that we did not perform any manual tuning of evaluation metrics. For each benchmark, we directly adopt the standard core metrics defined in the original works [1, 3, 7], and we never adjust or tune these metrics to reveal or amplify information leakage; all these metrics show information leakage in consistence. Regarding the reviewer’s concerns about generalizability and potential bias, we emphasize that leak@k is a meta-metric, which can be seamlessly applied to any base metric that based on generations.
>
> **Response:** I appreciate the clarification that leak@k evaluations were performed without manual tuning. However, I would like to note that leak@k being a meta-metric that inherits the properties of its underlying base metrics also inherits their limitations: its validity and applicability depend critically on (1) the answer formatting conventions of each dataset, and (2) the appropriateness of the generation-based metric chosen for the formatting. While I recognize that the experiments were conducted without manual tuning, the sensitivity of the metric to aspects such as answer formatting, which may vary even within the same dataset, remains a methodological weakness that should be considered in practice.
>
> ---
> > Our work performs a complete and systematic sweep across all widely used decoding configurations, varying temperature and top-p over the full standard ranges. We do not tune or select any particular (temperature, top-p) configuration.
>
> **Response:** Thank you for the clarification regarding their systematic evaluation across decoding configurations. The comprehensive sweep and the stability of utility metrics across settings are valuable. However, my concern about the practical interpretation of the leak@k metric for **establishing unlearning success** still remains. While the paper effectively demonstrates how leak@k can identify blind-spots in current evaluation practices, I believe additional guidance is needed on the following question: **Based on the leak@k metric, what constitutes a successful unlearning scenario, and under which parameter settings (temperature, top-p) should this be assessed?**
>
> For example, this becomes particularly salient when examining Figure 3, where leakage consistently increases with the number of generations. From a practitioner's perspective, this raises an important question: **What threshold for the number of generations should be considered sufficient to confidently claim that unlearning has been achieved?** Without clearer guidance on this point, it remains challenging to determine whether observed leakage represents a fundamental failure of the unlearning method or an artifact of generating an arbitrarily large number of samples.

---

### Official Review · Reviewer_h724 · 2025-11-02

**Soundness:** 2
**Presentation:** 3
**Contribution:** 3
**Rating:** 6
**Confidence:** 4

**Summary:**

This paper systematically demonstrates a critical vulnerability in current LLM unlearning techniques, arguing that their perceived effectiveness is an artifact of flawed evaluation protocols based on deterministic (greedy) decoding. The authors posit that these methods fail to achieve robust forgetting in practical scenarios utilizing probabilistic sampling, introducing a novel meta-metric, $leak@k$, to quantify the expected worst-case information leakage ($S_j$) across $k$ generations. Through a large-scale empirical study on standard benchmarks (TOFU, MUSE-News, WMDP), they show that $leak@k$ scores for state-of-the-art unlearning methods rise sharply with $k$, proving that sensitive information reliably resurfaces and concluding that current techniques offer merely an "illusion of forgetting" rather than true knowledge removal.

**Strengths:**

1. The paper is well-written; its logical structure and clear organization make the core arguments easy to follow and understand. Its primary and most significant contribution lies in the clear and timely identification of a critical vulnerability in current LLM unlearning evaluation. The authors rightly argue that the field's widespread reliance on deterministic (greedy) decoding fosters a misleading "illusion of forgetting," which obscures substantial risks present in real-world probabilistic sampling scenarios. I really thank the authors who tackle this important limitation of the existing LLM unlearning benchmark. Recognizing and articulating this gap is a major contribution, as it challenges the validity of prior claims and sets a new, more rigorous standard for evaluating robust unlearning.

2. To formally address this evaluation gap, the paper introduces $leak@k$, a novel and principled meta-metric that is both original and technically sound. By formalizing the evaluation as the expected worst-case leakage over $k$ samples, $leak@k := \mathbb{E}[\max_{1 \le j \le k} S_j]$, the authors provide a flexible and well-defined tool for the community. The quality of this contribution is further reinforced by the development of a low-variance unbiased estimator, demonstrating a high degree of statistical rigor.

3. The paper's empirical validation is of exceptional quality, marked by its comprehensive scope and clarity. The authors conduct a large-scale, systematic study across three diverse benchmarks (TOFU, MUSE-News, WMDP) and a full suite of state-of-the-art unlearning methods. The results are presented unambiguously and provide compelling, data-driven evidence that all evaluated techniques consistently fail under this more realistic probabilistic assessment, thereby strongly supporting the paper's central message.

**Weaknesses:**

While I strongly agree with the paper's major contribution, the following weaknesses make it difficult to evaluate the work more highly:

1. While the paper is generally well-structured and the prose is clear, the inconsistent use of citation commands (e.g., \cite vs. \citep) detracts from its professional polish. A thorough revision to ensure appropriate and consistent citation formatting throughout the manuscript is necessary to improve overall readability.

2. A significant weakness is the omission of several recent and highly relevant LLM unlearning analysis/algorithm papers, including numerous works from ICLR 2025 [1, 2, 3, 4]. I believe the authors should discuss with the existing LLM unlearning papers that pointed out the limitations in LLM unlearning [1,2]. Also, to make a compelling case, the paper's claims must be validated against a more diverse state of the art. For instance, methods like [3] and [4] reported exceptional performance on the TOFU benchmark, and [4] specifically demonstrated that using LoRA for unlearning offers significant advantages in both successful unlearning results and cost-efficiency. It is essential to investigate whether the probabilistic leakage issue identified by the authors persists even in these more advanced and robust unlearning techniques.

3. The paper's focus on the worst-case score (maximum leakage) is well-justified for safety and privacy. However, the analysis misses an opportunity to provide a more complete picture of the failure mode by also reporting the average score on the forget set (i.e., $\mathbb{E}[S_j]$) under probabilistic decoding. Presenting this average leakage alongside the $leak@k$ (worst-case) score would offer a more nuanced diagnostic, allowing a distinction between models that leak catastrophically but rarely versus those that leak more moderately but consistently. This additional statistic would enrich the understanding of how different unlearning methods fail.

[1] Catastrophic Failure of LLM Unlearning via Quantization, ICLR 2025

[2] Towards Effective Evaluations and Comparisons for LLM Unlearning Methods, ICLR 2025

[3] LLM Unlearning via Loss Adjustment with Only Forget Data, ICLR 2025

[4] Towards Robust and Parameter-Efficient Knowledge Unlearning for LLMs, ICLR 2025

**Questions:**

I have summarized my main weaknesses/questions in the Weaknesses section, so please refer to that content. Below are additional questions:

1. The proposed evaluation metric, which requires $n=200$ samples per prompt for a stable $\hat{leak@k}$ estimate, is necessarily more computationally expensive than the greedy-decoding baseline. Could the authors provide a quantitative analysis of this evaluation cost overhead? Discussing this practical trade-off would significantly strengthen the paper's assessment of the metric's viability for large-scale benchmarking.

2. The extensive experiments compellingly demonstrate that all evaluated unlearning methods remain vulnerable to probabilistic leakage. Based on the insights gained from designing $leak@k$ and observing these failure modes, what advice or intuition can the authors offer for designing future unlearning algorithms? Specifically, what properties should a truly robust algorithm possess to succeed under this more realistic evaluation? A discussion on this would be highly valuable for guiding future research in the field.

---

> ### Author Response · Authors · 2025-11-30
>
> > While the paper is generally well-structured and the prose is clear, the inconsistent use of citation commands (e.g., \cite vs. \citep) detracts from its professional polish. A thorough revision to ensure appropriate and consistent citation formatting throughout the manuscript is necessary to improve overall readability.
>
> **Response:** We thank the reveiwer for the comment. We adjusted the citation formatting in the revision.
>
>
> > A significant weakness is the omission of several recent and highly relevant LLM unlearning analysis/algorithm papers, including numerous works from ICLR 2025 [1**, 2**, 3**, 4**]. I believe the authors should discuss with the existing LLM unlearning papers that pointed out the limitations in LLM unlearning [1**,2**]. Also, to make a compelling case, the paper's claims must be validated against a more diverse state of the art. For instance, methods like [3**] and [4**] reported exceptional performance on the TOFU benchmark, and [4**] specifically demonstrated that using LoRA for unlearning offers significant advantages in both successful unlearning results and cost-efficiency. It is essential to investigate whether the probabilistic leakage issue identified by the authors persists even in these more advanced and robust unlearning techniques.
> > [1**] Catastrophic Failure of LLM Unlearning via Quantization, ICLR 2025
> [2**] Towards Effective Evaluations and Comparisons for LLM Unlearning Methods, ICLR 2025
> [3**] LLM Unlearning via Loss Adjustment with Only Forget Data, ICLR 2025
> [4**] Towards Robust and Parameter-Efficient Knowledge Unlearning for LLMs, ICLR 2025
>
> **Response:** We appreciate the reviewer for the great suggestion to study more models using our metric. Below, we summarize the newly added models and evaluations for each benchmark; full plots and experimental details can be found in the revision.
>
> **TOFU.** We implemented the method from [4**], **LoUK**, on the TOFU benchmark using  LLaMA-3.2-1B. **Similar** to **all** other unlearned models, we observe that this method exhibits **clear** information **leakage** under relatively high-randomness decoding strategies. We provide heatmaps for each configuration in **Fig. 2**, **page 8** and **Fig. A1**, **page 19**. Below, we report some of the obtained results for **LoUK** model on TOFU dataset using **ES** as the core metric.
>
> | (temp., top-$p$)   | 1    | 2    | 4    | 8    | 16   | 32   | 64   | 128  |
> | -------- | ---- | ---- | ---- | ---- | ---- | ---- | ---- | ---- |
> | $(0.0,0.0)$ (greedy decoding)  | 0.015 | 0.02 | 0.02 | 0.02| 0.02 | 0.02 | 0.02 | 0.02
> | $(0.2,0.2)$ |  0.03    |   0.03   |   0.03   | 0.03     | 0.03     | 0.03     |  0.03    | 0.03     |
> |   $(0.2,1.0)$       | 0.07     | 0.08     | 0.10     |  0.12    |  0.15    |  0.20    | 0.27     |  0.38   |
> | $(1.0,1.0)$  | 0.05 | 0.08  | 0.13  | 0.20  | 0.29  | 0.37  | 0.48  | 0.60 |
>
>
> We also conducted a new set of experiments on this dataset using a new core metric, **LLM-as-Judge (LJ)**, evaluated on all unlearned models under two sampling configurations: a low-randomness setting $(T,p)=(0.2,0.2)$ and a high-leakage setting $(T,p)=(1.0,1.0)$, reported in **Fig 3**, **page 8**. **Aligned** closely with the behavior observed using the **ES** metric in Fig. 2, page 8, the results obtained with **LJ** **confirm** that under high-randomness sampling, the unlearned model **exhibits** a clear risk of information leakage. In the following, we present the result of leak@k from some unlearned models using LJ metric under $(T,p)=(1.0,1.0)$ setting on TOFU dataset.
>
> | Method   | 1    | 2    | 4    | 8    | 16   | 32   | 64   | 128  |
> | -------- | ---- | ---- | ---- | ---- | ---- | ---- | ---- | ---- |
> | GradDiff  | 0.40 | 0.54 | 0.65 |0.74| 0.82 | 0.87| 0.92 | 0.95 |
> | NPO |  0.35    |   0.46   |   0.57   | 0.65     | 0.73     | 0.79    | 0.85   | 0.90    |
> |   BLUR-NPO       | 0.25    | 0.35  |0.45    |  0.40   |  0.62   |  0.69   | 0.74     |  0.79    |
> | LoUK  | 0.03  | 0.05  | 0.08  | 0.11  | 0.15  | 0.19  | 0.24  | 0.29 |

---

> ### Author Response · Authors · 2025-11-30
>
> **MUSE.** We implemented the proposed algorithm in [1**], **SURE**, using  LLaMA2-7B model and evaluated it on the MUSE benchmark with **RS** as the core metric. **Similar** to other models, we observe that this method also **fails** under our metric leak@k under relatively high-randomness decoding strategies. We provide heatmaps for each configuration in **Fig. 4**, **page 9** and **Fig. A3**, **page 20**. Below, we report some of the obtained results for **SURE** model.
>
>
> | (temp., top-$p$)   | 1    | 2    | 4    | 8    | 16   | 32   | 64   | 128  |
> | -------- | ---- | ---- | ---- | ---- | ---- | ---- | ---- | ---- |
> | $(0.0,0.0)$ (greedy decoding)  | 0.30 | 0.30 | 0.30 | 0.30 | 0.30 | 0.30 | 0.30 | 0.30 |
> | $(0.2,0.2)$ |  0.32    |   0.32   |   0.32   | 0.32     | 0.32     | 0.32    |  0.32    | 0.32     |
> |   $(0.2,1.0)$       | 0.33     | 0.43     | 0.54     |  0.64    |  0.72    |  0.80    | 0.86     |  0.90    |
> | $(1.0,1.0)$  | 0.25  | 0.37  | 0.50  | 0.62  | 0.73  | 0.82  | 0.89  | 0.94 |
>
>
>
> **WMDP.** We further extended our evaluation on WMDP by applying **LJ** to RMU model for two sampling configurations $(T,p)=(0.2,0.2)$ and $(T,p)=(1.0,1.0)$. **Similar** to the **Acc** core metric, RMU exhibits clear information leakage under high-randomness decoding setting $(T,p)=(1.0,1.0)$.
>
> | (temp., top-$p$)   | 1    | 2    | 4    | 8    | 16   | 32   | 64   | 128  |
> | -------- | ---- | ---- | ---- | ---- | ---- | ---- | ---- | ---- |
> | $(0.2,0.2)$  | 0.047 | 0.051 | 0.057 | 0.064| 0.071  | 0.080 | 0.091 | 0.108 |
> | $(1.0,1.0)$ |  0.037    |   0.052   |   0.068   | 0.086     | 0.107     | 0.127     |  0.145   | 0.162   |
>
>
> > The paper's focus on the worst-case score (maximum leakage) is well-justified for safety and privacy. However, the analysis misses an opportunity to provide a more complete picture of the failure mode by also reporting the average score on the forget set (i.e., $\mathbb{E}[S_j]$) under probabilistic decoding. Presenting this average leakage alongside the $leak@k$ (worst-case) score would offer a more nuanced diagnostic, allowing a distinction between models that leak catastrophically but rarely versus those that leak more moderately but consistently. This additional statistic would enrich the understanding of how different unlearning methods fail.
>
> **Response:** We thank the reviewer for the valuable suggestion. We included the average leakage $\mathbb{E}[S_j]$ in the revision. However, we find that **average leakage is not informative for robustness**. As the table below shows, the numerical difference between greedy decoding and probabilistic average leakage is consistently small, and in some cases the average under probabilistic decoding does not exceed greedy decoding. This experiment is conducted on the TOFU dataset using ES core metric, where leakage is identified by ES = 1 and no leakage by ES = 0. The average leakage $\mathbb{E}[S_j]$ is computed as the proportion of responses exhibiting leakage over the total number of sampled responses.
>
>
> |         Model         | Original | Retrain | NPO  | BLUR-NPO | GradDiff | RMU  | SimNPO | NPO+ENT    |
> |:---------------------:|:--------:|:-------:|:----:|:--------:| -------- | ---- | ------ | --- |
> |    Greey Decoding     |   0.343   |   0.083   | 0.19 |   0.095    | 0.295     | 0.315 | 0.275   |  0.252   |
> | $(T,p)=(1.0,1.0)$ |   0.308   |  0.168   | 0.204 |   0.111   | 0.261     | 0.285 | 0.252   | 0.246    |
>
>
> This occurs because probabilistic decoding induces a **redistribution** of leakage rather than a **uniform** increase: some queries that are *safe* under *greedy decoding* begin *leaking* when sampled, while some queries that *leak* under greedy decoding occasionally produce *safe* outputs. These opposing effects *cancel out* in the *mean*.
>
>
> To illustrate this effect, we split the 400 questions of the forget-set into **two** groups based on greedy decoding: those that are safe (19%) and those that are unsafe (81%). When switching to probabilistic sampling, questions that were previously safe begin to exhibit new leakage, their average score  $\mathbb{E}[S_j]$ increases from *0 to 0.107*. In contrast, questions that were previously unsafe become partially safer on average, with their $\mathbb{E}[S_j]$ decreasing from an implied *1.0 to 0.624*. These opposite shifts largely cancel each other out, resulting in a nearly unchanged overall average leakage. This demonstrates that the mean leakage alone can be misleading, as it masks substantial changes in model behavior under probabilistic decoding.
>
> |                   | Unsafe Questions | Safe Questions |
> |:-----------------:|:--------------:|:----------------:|
> | $\mathbb{E}[S_j]$ |     $$1\rightarrow 0.624$$      |      $$0\rightarrow0.107$$       |

---

> > ### Author Response · Authors · 2025-11-30
> >
> > > The proposed evaluation metric, which requires $n=200$ samples per prompt for a stable $\hat{leak@k}$ estimate, is necessarily more computationally expensive than the greedy-decoding baseline. Could the authors provide a quantitative analysis of this evaluation cost overhead? Discussing this practical trade-off would significantly strengthen the paper's assessment of the metric's viability for large-scale benchmarking.
> >
> > **Response:** We thank the reviewer for the suggestion. We added a discussion on the evaluation-time in **pages** **18** and **19**, summarized in the table below.
> >
> > |  Benchmark  | Time for Generation | Time for Evaluation RS/ES |
> > |:-----------:|:-------------------:|:------------------------------:|
> > | TOFU |        10min        |             30min           |
> > |    MUSE     |        1h             | 15min                               |
> > |    WMDP     |        1h             |       15min    |
> >
> >
> > **Table.** Across all three benchmarks, we generate $n=200$ model responses per prompt. The TOFU, MUSE, and WMDP datasets contain 400, 100, and 200 prompts, respectively. “Generation time” is the total time to produce all responses for the full prompt set. “Evaluation time” is the time to compute ES/RS leakage scores for all generated responses. All generation and ES/RS Evluation steps are done by using one or two NVIDIA A40 GPUs.
> >
> >
> > We note that in modern LLM evaluation, multi-sample evaluation is standard practice. **Pass@k** is widely used in **SWE**, **HumanEval**, and **GSM8K** [11, 12] precisely because multiple generations are essential for capturing the output distribution and obtaining stable performance estimates. Our metric leak@k follows the **same** principle.
> >
> > > The extensive experiments compellingly demonstrate that all evaluated unlearning methods remain vulnerable to probabilistic leakage. Based on the insights gained from designing $\hat{leak@k}$ and observing these failure modes, what advice or intuition can the authors offer for designing future unlearning algorithms? Specifically, what properties should a truly robust algorithm possess to succeed under this more realistic evaluation? A discussion on this would be highly valuable for guiding future research in the field.
> >
> > **Response:** We thank the reviewer for the comment. We have introduced a meothd, **NPO-Fix,** *integrating generation directly* into the unlearning process. This method leverages sampling to automatically surface these hidden leakage modes, expands the forget set with the leaked generations, and then iteratively refines the model on this augmented data. NPO-Fix achieves **substantially better performance** than existing methods. Our dynamic dataset augmentation approach is a promising
> > approach **incorporating generation into the unlearning stage**. We provided more details in **Appendix F**.

---

### Meta-Review · Area_Chair_7Gvr · 2025-12-04

**Summary:**

The paper identifies that "unlearned" language models actually leak the supposedly-unlearned information if one considers sampling top k rather than doing greedy decoding. The authors show that this failure appears in several benchmarks and baselines. The empirical evaluation was extended during the rebuttal period. The observation reported in the paper is practically relevant. However, several concerns remain post rebuttal regarding depth and conceptual framing. In particular, the authors argue that the retrain from scratch model is not the gold standard because it sometimes leaks information in benchmarks like MUSE. This was in response to reviewer concerns (which had issues, I acknowledge and fully take that into consideration). I would argue that the authors are conflating here the definition of unlearning, and the quality of the benchmarks. If the retrain model leaks, this would suggest the benchmark itself contains duplicates/similar information in the data that was in the retain set. This does not change the fact that the retrain model represents the upper bound of unlearning achievable by the architecture on that data distribution. The authors are welcome to clearly define what they mean by unlearning, but they do not do so. Also, I would encourage the authors to design a clean benchmark where they would know the retrain will achieve exactly what they want. It is also important to discuss limits as k goes to infinity. Will their unlearning be satisfied?

On a related note, the paper argues that leakage increases with k. However, as k increases, the probability of generating any specific tail-end distribution increases. It would be great to see a trade-off between k and general utility.

Overall, for a paper that is pushing for a standard evaluation metric to be adopted by the community, I would need to see more clarity in definitions and problem setup, and clear discussions around the meaning of k.

**Reviewer Concerns:**

The key concerns addressed were around additional benchmarks and baselines. The authors also added additional metrics, and different reporting of results (average vs worst case), as well as quantified computational cost. Remaining concerns: novelty, retrain from scratch baseline (gold standard or not?), evaluation on larger models (minor, in my opinion).

**Reviewer Scores:**

Reviewer A144 and Reviewer VUzu - unlikely to change. One of them already responded and were sticking to their points. Also note that the reviews were likely LLM generated.

Reviewer h724 - maintain or increase. they asked for additional baselines, which the authors provided.

Reviewer RrXa - primary weakness around the lack of baselines and benchmarks was (partially) addressed.

---

### Decision · Program_Chairs · 2026-01-26

Reject